# LTMD: Learning Improvement of Spiking Neural Networks with Learnable Thresholding Neurons and Moderate Dropout

**Siqi Wang**[*]
Electrical and Electronic Engineering
Nanyang Technological University
Singapore
siqi002@e.ntu.edu.sg

**Tee Hiang Cheng**
Electrical and Electronic Engineering
Nanyang Technological University
Singapore
ethcheng@ntu.edu.sg

**Meng-Hiot Lim**
Electrical and Electronic Engineering
Nanyang Technological University
Singapore
emhlim@ntu.edu.sg

## Abstract

Spiking Neural Networks (SNNs) have shown substantial promise in processing spatio-temporal data, mimicking biological neuronal mechanisms, and saving computational power. However, most SNNs use fixed model regardless of their locations in the network. This limits SNNs' capability of transmitting precise information in the network, which becomes worse for deeper SNNs. Some researchers try to use specified parametric models in different network layers or regions, but most still use preset or suboptimal parameters. Inspired by the neuroscience observation that different neuronal mechanisms exist in disparate brain regions, we propose a new spiking neuronal mechanism, named learnable thresholding, to address this issue. Utilizing learnable threshold values, learnable thresholding enables flexible neuronal mechanisms across layers, proper information flow within the network, and fast network convergence. In addition, we propose a moderate dropout method to serve as an enhancement technique to minimize inconsistencies between independent dropout runs. Finally, we evaluate the robustness of the proposed learnable thresholding and moderate dropout for image classification with different initial thresholds for various types of datasets. Our proposed methods produce superior results compared to other approaches for almost all datasets with fewer timesteps. Our codes are available at `https://github.com/sq117/LTMD.git`.

## 1 Introduction

Modelled after the impulse communication between biological neurons, spiking neural network (SNN) is a new class of neural network with neurons exhibiting a distinctive binary output property. SNNs have shown great potential in event-driven data processing, computation reduction, and network biological plausibility enhancement. However, unoptimized learning algorithm and complex neuronal dynamics make it challenging to construct high-performance SNN models.

Current SNN training approaches can generally be divided into two categories: ANN-to-SNN conversion and SNN direct training algorithms based on gradient descent. Both ways only focus

---

[*]corresponding author

36th Conference on Neural Information Processing Systems (NeurIPS 2022).

on adjusting synaptic connections to minimize the error between a model's output and target values with the same neuronal model being used across the whole network. However, it has been found from neuroscience observations that biological neurons' dynamics vary with their relative locations in the brain especially for primates [1, 2]. Therefore, using the same neuronal model across the whole SNN may limit the SNN's expression ability as compared to allowing heterogenous neuronal models to be used in the same SNN. Although some researchers have proposed many techniques to balance presynaptic inputs and neuronal behaviors, for example, threshold regularization [3] and spike-based normalization [4], they are either targeting at neuronal state distribution or only applicable for ANN-to-SNN conversion algorithms.

Dropout [5, 6] is a commonly used regularization methodology to train neural networks. As dropping neurons are randomly selected, each run of network will lead to the input interacting with different sub-models, which will cause the results in different runs to deviate unpredictably. Consequently, a regularization technique is needed to minimize the inconsistencies between output probability distributions during the training phase to improve the performance.

In this paper, we propose a new learnable thresholding mechanism with a moderate dropout method to enhance the learning of modulated SNNs. The learnable thresholding mechanism can be integrated with the backpropagation-based SNN direct training algorithm so that neuronal parameters can be changed dynamically and self-optimize during training. We adapt the DenseNet architecture and modify the encoding and decoding layers to make the network suitable for diverse datasets. A new 'moderate dropout' technique is developed to minimize inconsistencies between network sub-models generated in different runs. We demonstrate the proposed methods can enhance the SNN's stability and performance through evaluation based on static and neuromorphic datasets. To the best of our knowledge, our proposed SNNs achieve the best or comparable accuracies for all the datasets tested compared to other state-of-the-art SNNs.

## 2 Related Work

### 2.1 ANN-to-SNN Conversion

ANN-to-SNN conversion [7, 4, 8] aims to convert trained ANNs to SNNs by using rate coding scheme instead of ReLU activation to represent data flowing inside the network [9]. An ANN of the same structure will be trained first, followed by adjusting the trained synapses for spatio-temporal data. Many constraints such as bias term needs to be excluded, and only average pooling should be used, etc. need to be incorporated in the pre-training [10]. However, this indirect training approach poses a lot of problems, such as temporal information is disregarded during ANN training, long delay incurred for encoding presynaptic inputs, etc., which seriously limits the network's application spectrum. Furthermore, this training method is inappropriate for neuromorphic datasets since only spatial data can be utilized.

### 2.2 Direct-Trained Deep SNNs and Neuronal Model

The other method is to train SNNs directly based on gradient descent by backpropagating errors [11] in both spatial and temporal dimensions. Various strategies have been proposed to realize gradient value calculation for non-differentiable spike activation. Some recent works apply pseudo derivative algorithm to replace the non-differentiable spike triggering part of membrane potential curves with an auxiliary function and report satisfactory results [12, 13, 14]. As both spatial and temporal information are used in network training, dataset restrictions will not blight direct training method as in the ANN-to-SNN conversion approach. In addition, simulation latency can be significantly shortened because more efficient data encoding schemes can be freely used with gradient descent.

The direct supervised learning methods train SNNs based on event-driven data and gradient descent, which is scalable and thus suitable for deep SNNs. In [15, 10, 16], several approaches to build deep SNN with direct learning methods are described. It is pointed out in [9] that information expressiveness loss attributes to step functional activation causes deep SNNs to suffer performance degradation. Multiple deep network architectures such as VGG [17], ResNet [18, 9], and DenseNet [19] have been leveraged to SNNs so that spiking neurons can maintain sufficient information for transmission.

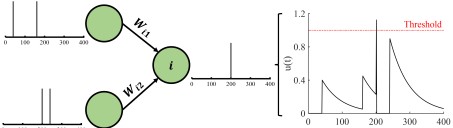 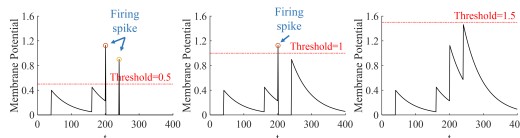

(a) Information transmission between neurons.   (b) Potential responses with different thresholds.

Figure 1: Illustration of (a) spiking neuron membrane potential update in LIF model. (b) neuron responses to the same input when using different threshold values.

Spiking neuronal model plays a crucial role in controlling data transmission in SNNs. Inspired by biological neurons, some researchers add lateral effect to spiking neurons [20, 13]. Some use parametric neuronal models for different tasks by making time constant to be learnable [21, 22, 23]. One group proposes a network which contains neurons of adaptive thresholds [24], and another group makes the potential leaky process to be learnable [25]. Better accuracies and higher network robustness are demonstrated with these neuronal model improvements.

### 2.3 Dropout

One critical issue in using backpropagation to update synaptic connections is the gradient vanishing and exploding problem especially for deep networks. To reduce the degradation effect, numerous optimization techniques such as regularization [3, 6] and normalization [4, 16] have been designed. Dropout is one of the widely applied regularization methods to prevent overfitting in deep networks. Neurons will be retained randomly with certain probability and form multiple sub-models for training, after which the trained network will be generated based on all these sub-models. Some researchers modify dropout to make it suitable for spatio-temporal data and successfully apply it on SNNs. However, as spiking neurons have both spatial and temporal connections, this results in sub-models to exhibit high level of inconsistencies in different sub-model runs.

## 3 Methods

In this section, we introduce the learnable thresholding mechanism and moderate dropout and expound the proposed SNN.

### 3.1 Leaky Integrate-and-Fire Model with Learnable Thresholding

Spiking neurons are the basic elements for information processing and transmission via alternating the membrane states in SNNs. Many neuronal models have been proposed by neuroscientists to simulate biological neurons' behaviors on computers. Due to the complexity of neuronal dynamics in real nervous systems, there exists trade-offs between biological plausibility and computational cost in computer simulation. The Leaky Integrate-and-Fire (LIF) is a simple and pre-eminent mathematical model for modelling such neuronal behaviors as potential update, spike emission, and state reset. It can be described as

$$\tau \frac{du(t)}{dt} = -u(t) + I(t) \tag{1}$$

where $\tau$ is a time constant which represents how fast a neuron's potential will decay with time, $u(t)$ and $I(t)$ are the membrane potential and integrated presynaptic input at time $t$, respectively. In this work, we set neurons' initial states and reset potentials to be zero. Fig. 1a illustrates a neuron's different behaviors under presynaptic inputs.

To suit computer simulation, we need to temporally discretize the LIF model and spiking neurons' states. The following numerical representation of a spiking neuron's potential states can be derived from Eq. 1:

$$u_i^{t,n} = \kappa u_i^{t-1,n}(1 - o_i^{t-1,n}) + \sum_{j \in l(n-1)} W_{ij} o_j^{t,n-1} \tag{2}$$

$$o_i^{t,n} = f(u_i^{t,n}) \tag{3}$$

where $u_i^{t,n}$ is the membrane potential of postsynaptic neuron $i$ located in layer $n$ at time $t$, $\kappa$ is the decay factor, $o_j^{t,n-1}$ is the output spike from presynaptic neuron $j$, and $l(n-1)$ represents all presynaptic neurons connected to target $i$. $f(u)$ is the step functional activation to determine if the target neuron will fire at current timestep, and $f(u) = 1$ when $u \geq V_{th}$, otherwise $f(u) = 0$. Eq. 2 describes the membrane potential update and Eq. 3 introduces how spiking neurons emit new spikes. In the LIF model, a postsynaptic neuron's state in the previous timestep and presynaptic inputs from preceding layers spatio-temporally govern the target neuron's behaviors.

As SNNs need to use backpropagation to update their parameters, we need to circumvent the obstacle of nondifferentiable activation function $f(u)$. Hence, we adopt the idea of pseudo derivative and designate an auxiliary rectangular function to serve as the approximated gradient:

$$f'(u) = \begin{cases} 1, & |u - V_{th}| < 0.5 \\ 0, & otherwise \end{cases} \tag{4}$$

By doing so, our SNNs is endowed with the parameter optimization capability through backpropagation.

Our learnable thresholding mechanism is developed on the basis of LIF model by specifically considering the diversities among spiking neurons even if they are in the same network. This means a neuron's response depends not only on its internal state but also the threshold level. In order to investigate the effects of different threshold values on neuronal behaviors, we assume that two presynaptic neurons are connected with one postsynaptic neuron through synapses of weight $w_1$ and $w_2$, respectively. When the postsynaptic neuron receives input spikes from preceding layers, its internal state will change accordingly based on the weighted input spike trains and the preset thresholds. As illustrated in Fig. 1a, two presynaptic neurons are triggered at $t = \{40, 160\}$ and $\{200, 240\}$ respectively. According to Eq. 2, the weighted input to postsynaptic neuron is $I(t) = w_1(\delta(t-40) + \delta(t-160)) + w_2(\delta(t-200) + \delta(t-240))$, where $\delta(t)$ represents the Dirac delta function. Fig. 1b depicts the postsynaptic neurons' responses to the same input $I(t)$ when using different threshold values, showing a trend of decreasing output spike number and lower system sensitivity to the external stimulations as threshold value becomes higher. The high sensitivity to the input helps the system to track small and instant input signals while low sensitivity can enhance the system's noise tolerance. Thus, considering the effects of various threshold levels on spiking neurons' behaviors, our learnable thresholding mechanism can achieve optimal sensitivity using the following modified neuron response function:

$$o_i^{t,n} = g(u_i^{t,n}, M^n) \tag{5}$$

where $M^n$ is the threshold value for neurons in the $n$th layer. $g(u, m) = f(u - m)$, and $f(x)$ is the same step function with zero valued threshold and follows Eq. 4.

The main idea of learnable thresholding is to analytically find the comprehensive gradient of the loss function, $\nabla \mathcal{L}(W, M) = [\frac{\partial \mathcal{L}}{\partial W}, \frac{\partial \mathcal{L}}{\partial M}]$, and then simultaneously update synaptic connections $W$ and the neuron's threshold $M$ until convergence. From Eq. 2, 4, and 5, partial derivatives of the loss function $\frac{\partial \mathcal{L}}{\partial W}$ and $\frac{\partial \mathcal{L}}{\partial M}$ can be calculated by

$$\frac{\partial \mathcal{L}}{\partial W^n} = \sum_{t=1}^{T} \frac{\partial \mathcal{L}}{\partial o^{t,n}} \frac{\partial o^{t,n}}{\partial u^{t,n}} \frac{\partial u^{t,n}}{\partial W^n} \tag{6}$$

$$\frac{\partial \mathcal{L}}{\partial M^n} = \sum_{t=1}^{T} \frac{\partial \mathcal{L}}{\partial o^{t,n}} \frac{\partial o^{t,n}}{\partial M^n} \tag{7}$$

The term $\frac{\partial o^{t,n}}{\partial M^n}$ in Eq. 7 represents the derivative of a neuron's output with respect to its threshold. It can be shown that $\frac{\partial o^{t,n}}{\partial M^n} = \frac{\partial f(u^{t,n} - M^n)}{\partial M^n} = -f'(u^{t,n} - M^n)$, whose value is determined by the auxiliary function's gradient defined in Eq. 4. It can be seen from the above formulae that there is no restriction on the threshold value $M$. If the threshold becomes extremely low or high, a neuron will keep firing or stay silent when receiving the input from presynaptic neurons, causing difficulties in transmitting information. To ensure threshold values to stay within an appropriate region, we create a new parameter $k$ to define $M$ using hyperbolic tangent relation, formulated as $M = \tanh(k)$. This stabilizes neuronal activity by avoiding having too many or too few neurons to fire due to extreme

threshold levels. With this, Eq. 7 can be expressed as:

$$\frac{\partial \mathcal{L}}{\partial k^n} = \sum_{t=1}^{T} \frac{\partial \mathcal{L}}{\partial o^{t,n}} \frac{\partial o^{t,n}}{\partial M^n} \frac{\partial M^n}{\partial k^n} = -\sum_{t=1}^{T} \frac{\partial \mathcal{L}}{\partial o^{t,n}} f'\big(u^{t,n} - \tanh(k^n)\big) \operatorname{sech}^2(k^n) \qquad (8)$$

With this enhancement, a neuron's threshold $M$ can be trained iteratively using the backpropagation method and takes a value within the range $(-1, 1)$. Furthermore, the gradient of the hyperbolic tangent function becomes very small when $k$ becomes infinitely large or small, ensuring that there will not be any sudden big change on the threshold value and thus achieves the stabilization effect. Although some recent works have treated threshold level as variable to introduce heterogeneity to SNN, grim prerequisites such as neuronal states transformation [26] or pre-defined firing count [27] are required. Some other STDP-based approaches only work on excitatory neurons [28], leading to slow convergence and heavy computational load to individually update every neuron's threshold. However, the proposed strategy provides an easy approach to grant neuronal thresholds with learning capability under light computational load and can be freely migrated to other gradient descent-based SNNs, which is exactly the aim of this paper.

Though setting a lower threshold level may have a similar effect as having higher synaptic weights, changing threshold values has other benefits, for example, the potential decay curve will not be stretched and the SNN will not overly depend on a subset of synapses for optimization. In this work, we apply the learnable thresholding mechanism to all neurons and let neurons in the same layer share the same threshold to reduce total learnable parameters incurred.

### 3.2 Input Encoding and Network Architecture

Since SNNs are inherently suitable for processing event-driven data, spike trains in neuromorphic datasets can be directly fed into SNNs without barrier. However, the situation becomes challenging for static datasets due to lack of temporal information. Rate coding scheme is widely applied in ANN-to-SNN conversion and some recent works [29, 6] aim to encode input into serial spikes generated with certain firing probability that is proportional to the original value. One spiking neuron can encode maximally $T + 1$ values into distinguishable spike trains of firing probabilities $\{0, \frac{1}{T}, \ldots, 1\}$ within simulation window $T$. Therefore, the rate coding scheme suffers long latency due to the inevitably long simulation period for maintaining high precision among inputs of a wide range. In this work, the first layer serves as the encoding layer that directly receives input data and then, converts data into spike signals before transmitting to the second layer. In other words, there is no value-to-spike pre-processing step, inputs from dataset can be liberally fed into the network even if they are not in spike forms. In this way, we can significantly shorten the simulation window, which makes it possible to apply our SNN for analysis within several timesteps.

DenseNet is proposed in [19] where network layers are densely connected so that data from preceding layers can be transmitted to subsequent layers directly through dense connections between neurons. In this work, we leverage the idea and build our SNNs with DenseNet structure to retain information from degradation caused by the step activation. Our proposed learnable thresholding method is applied to all neurons in the network and the normal dropout in fully-connected layers is replaced by our moderate dropout. A detailed structural illustration of our network is shown in Fig. 2.

### 3.3 Output Decoding and Lateral Interaction

In some works, the first firing neuron in the output layer indicates the predicted result of the network. However, as neurons' states are temporally discretized in simulation, there exists a situation in which more than one output neurons will fire together at the same timestep. To avoid such a situation, we choose to use a neuron's potential value instead of firing state for prediction. In this manner, a network will choose the neuron with the highest internal state as output even if multiple neurons are triggered simultaneously.

A neuron's internal state is determined by both environmental stimulus and local effects of its neighboring neurons. Excitatory and inhibitory interactions within the receptive field are termed as the lateral interaction property of neurons. Lateral interaction was first introduced to explain an optical phenomenon that color contrast will be exaggerated to form simultaneous contrast by visual systems when two color blocks of slightly different gray levels start to connect with each other. Lateral interactions can avoid information overload by dampening input from some neurons and enhancing

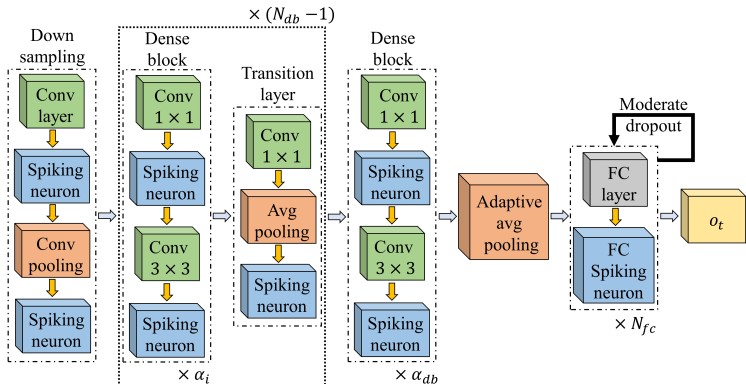

Figure 2: Network architecture of the proposed SNNs. The schematic represents data transmission in one timestep, $N_{fc}$, $N_{db}$, and $\alpha_i(\alpha_{db})$ are the number of fully-connected layers, total dense block number, and layer iterations in each dense block.

input from others. In this respect, it is used to sharpen the image and enhance system noise tolerance capability. In this work, we consider the lateral interactions among neurons in fully-connected layers to enhance system robustness and plausibility. A neuron's state is dominated by two components: potential update triggered by previous state or preceding neurons, and interaction with its neighboring neurons in the same layer. Membrane potential of spiking neurons in fully-connected layers and neuron's output can be derived from Eq. 2 and 3 as follows:

$$u_i^{t,n} = \kappa u_i^{t-1,n}\big(1 - f(u_i^{t-1,n})\big) + \sum_{j \in l_{fc}(n-1)} W_{ij} o_j^{t,n-1} + W_{rf} o_{rf}^{t-1,n} \tag{9}$$

$$o^{t,n} = \begin{cases} u^{t,n}, & n \in l_{fc} \\ f(u^{t,n}), & otherwise \end{cases} \tag{10}$$

where the term $W_{rf} o_{rf}^{t-1,n}$ represents lateral interaction between the target neuron and its surrounding neurons, and $l_{fc}(n-1)$ is a set of neurons connected to the target neuron $i$ in fully-connected layer $(n-1)$. $W_{rf}$ is a matrix of lateral interaction learnable weights, whose size defines the width of the receptive field. As aforementioned, the proposed network uses potential or spike as output in different layers, following Eq. 10. By integrating the lateral interaction mechanism, the robustness of the network will be ameliorated because the noise effect vanishes when neuron interactions in a larger area are considered, especially for neuromorphic datasets that suffer from greater noise disturbances.

### 3.4 Moderate Dropout

Noting that highly deviatory sub-models generated by dropout will be integrated and scaled, which may inhibit the constructed SNN's performance. In our SNN, the last-layer neurons' time-averaged potentials can be treated as the probabilities of classifying input pattern to each potential class. As such, we propose the moderate dropout that aims to minimize inconsistencies between different sub-models' output probability distributions. The Kullback-Leibler (KL) divergence score is used to measure the inconsistence in the output probability distributions of different network runs as illustrated in Fig. 3. With $P$ and $Q$ being the averaged probability distributions over the entire simulation period on the same probability space, KL divergence is defined as

$$D_{KL}(P \parallel Q) = \sum_x P(x) \log\left(\frac{P(x)}{Q(x)}\right) \tag{11}$$

We compare sub-models' output distributions and incorporate the divergence into the overall loss of the network to minimize the inconsistencies between output distributions. Therefore, the overall loss function comprises the loss of actual output distributions against the target labels and the KL divergence loss between different distributions output by the same model with moderate dropout.

It can be observed from Eq. 11 that the KL divergence, depicting the relative entropy of $P$ with respect $Q$, is asymmetric in nature, such that $D_{KL}(P \parallel Q) \neq D_{KL}(Q \parallel P)$. In this respect, we involve

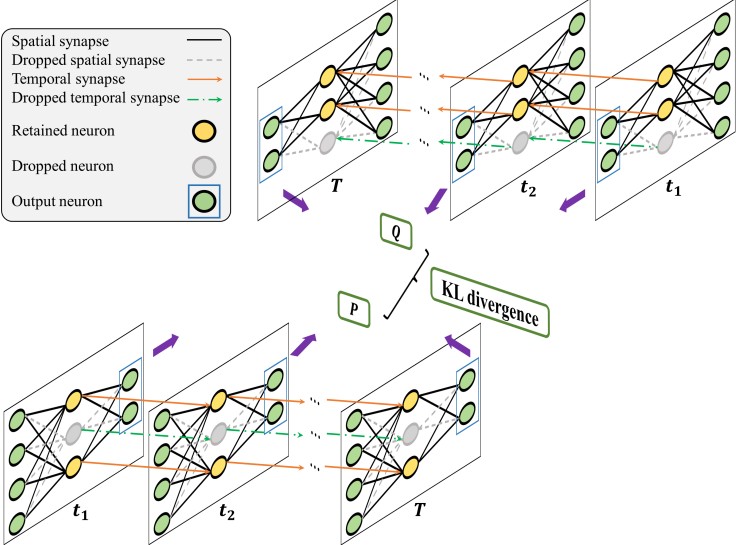

Figure 3: Illustration of KL divergence between different runs of SNN sub-models. Each row of frames represents one sub-model's temporally unfolded architecture after neuron dropping. $P$ and $Q$ are two averaged sub-model output distributions over the entire simulation period $T$.

both divergences in the network loss calculation. Furthermore, we use one additional parameter $\gamma$ to control the impact of moderate dropout in the overall loss function as follows:

$$\mathcal{L}_{overall} = \frac{1}{2}(\mathcal{L}_P + \mathcal{L}_Q) + \gamma\big(D_{KL}(P \parallel Q) + D_{KL}(Q \parallel P)\big) \tag{12}$$

From the discussion on dropout, it can be seen that the probability $p$ in which neurons are obscured from the fully-connected layers are stipulated before training and will be kept fixed throughout the training phase when drawing from the Bernoulli distribution. However, this does not take into account the membrane potential values of the neurons before deciding to drop from the network, leading to possible scenarios where essential neurons with high potential are temporarily eliminated from the network. Especially in our SNNs, the potential values are directly used to represent the neurons' states in fully-connected layers. The dropping probability of neurons should be accorded some flexibility for alteration taking into consideration the neuronal membrane potential values so that neurons processing larger membrane values can have smaller dropping probabilities. If a neuron $i$ in fully-connected layer $n$ has a membrane value of $u_i^n$, and $u_i^n$ lies in a distribution with $u_i^n \in (u_{min}^n, u_{max}^n)$, where $u_{min}^n$ and $u_{max}^n$ correspond to the minimum and maximum potential values of neurons in that layer respectively, we can derive the following equation to transform and scale up the potential value $u_i^n$ and use it to compute the corresponding dropping probability $p_i^n$.

$$p_i^n = p_{drop} \cdot \frac{u_{max}^n - u_i^n}{u_{max}^n - u_{mean}^n} \tag{13}$$

Eq. 13 ensures that a neuron's transformed dropping probability is inversely proportional to its membrane potential and the average dropping probability of all neurons in the same layer equals to the drop rate $p_{drop}$, which has been defined before training. In this regard, neuron dropping probabilities can be altered throughout the training phase, with larger potentials denoting smaller probabilities, rather than being preset before training in conventional dropout. With the creation of potential-dependent probability tensor, this can be subsequently placed into the Bernoulli distribution to generate binary masks for different sub-models. KL divergence between the different output distributions from sub-model runs is incorporated into the loss function according to Eq. 12, with the overall objective being to minimize the loss, leading to increased consistencies among different runs of the network.

Table 1: SNN structural parameters.

| Dataset | $N_{db}$ | $\alpha_1$ | $\alpha_2$ | $\alpha_3$ | $\alpha_4$ | $N_{fc}$ |
|---|---|---|---|---|---|---|
| Static | 3 | 12 | 12 | 24 | - | 2 |
| Neuromorphic | 4 | 6 | 12 | 24 | 16 | 2 |

Table 2: Classification accuracies comparison among proposed methods and state-of-the-art records on various datasets. Number of timesteps and the highest accuracies achieved are highlighted in italics (in parentheses) and bold.

| Model | Method | MNIST | CIFAR10 | N-MNIST | DVS-CIFAR10 |
|---|---|---|---|---|---|
| [31] | ANN-to-SNN | 98.37% | 82.95% | - | - |
| [7] | ANN-to-SNN | - | 77.43% | - | - |
| [32] | ANN-to-SNN | - | - | 95.72% | - |
| [33] | ANN-to-SNN | - | 89.32% | - | - |
| [34] | ANN-to-SNN | 99.10% | - | - | - |
| [35] | ANN-to-SNN | 99.44% | 88.82% | - | - |
| [4] | ANN-to-SNN | - | 91.55% | - | - |
| [36] | ANN-to-SNN | - | 93.63% | - | - |
| [37] | Random Forest | - | - | - | 31.00% |
| [38] | SKIM | - | - | 92.87% (*360*) | - |
| [39] | HATS | - | - | - | 52.40% |
| [40] | DART | - | - | - | 65.78% |
| [41] | Streaming rollout ANN | - | - | - | 66.75% |
| [3] | Direct training | - | - | 98.74% (*300*) | - |
| [15] | Hybrid direct training | 99.28% (*175*) | - | - | - |
| [12] | Direct training | 99.42% (*30*) | 50.70% (*30*) | 98.78% (*30*) | - |
| [42] | Hybrid direct training | 99.49% (*400*) | - | 98.88% (*500*) | - |
| [30] | Direct training | **99.62%** (*400*) | - | - | - |
| [10] | Direct training | - | 90.53% (*12*) | 99.53% (*8*) | 60.50% (*8*) |
| [16] | Direct training | - | 93.16% (*6*) | - | 67.80% (*10*) |
| [6] | Direct training | 99.59% (*50*) | 90.95% (*100*) | 99.09% (*100*) | - |
| [13] | Direct training | 99.50% (*20*) | - | 99.45% (*20*) | - |
| [43] | Direct training | 99.46% (*25*) | - | 99.39% (*25*) | - |
| [44] | Direct training | - | - | 96.30% (*120*) | 32.2% (*80*) |
| [9] | Direct training | - | - | - | 70.2% (*8*) |
| LTMD | Direct training | 99.60% (*4*) | **94.19%** (*4*) | **99.65%** (*15*) | **73.30%** (*7*) |

# 4 Experiment

We evaluate our proposed learnable thresholding and moderate dropout by testing our deep SNNs of DenseNet architecture for classification tasks using both static and neuromorphic datasets. We demonstrate the superiority of our methods in terms of inference accuracies and simulation timesteps by comparing it with other state-of-the-art SNN approaches.

## 4.1 Empirical Evaluation

Due to the simplicity of MNIST, we specially create an 5-layer fully-connected SNN (*input-conv1-pool1-conv2-pool2-conv3-pool3-fc1-fc2-output*) for this dataset. All the other datasets are tested with deep-layer SNNs of DenseNet architecture. Detailed structures of the DenseNet SNNs we build are listed in Tab. 1.

We implement the proposed methods in our deep SNNs and test their accuracies for classification tasks on both static MNIST, CIFAR10 datasets, and neuromorphic N-MNIST, DVS-CIFAR10 datasets. We set the initial threshold parameter $k = 0.5$ and moderate dropout impact factor $\gamma = 0.5$ for all the datasets. Dataset description, network parameters, and setting details are given in Appendix. The results we obtained are shown in Tab. 2, it can be seen that our SNNs outperforms other state-of-the-art SNNs for all datasets except MNIST. Although our inference accuracy for MNIST is slightly lower than [30], our SNN achieves a comparable result with much fewer timesteps needed. This makes our proposed SNN suitable for fast data analysis.

Table 3: Inference results using proposed methods.

| Dataset | LIF | LT | LTMD |
|---|---|---|---|
| MNIST ($step = 4$) | 99.53% | 99.57% | 99.60% |
| CIFAR10 ($step = 2$) | 92.88% | 93.51% | 93.75% |
| N-MNIST ($step = 15$) | 99.55% | 99.58% | 99.65% |
| DVS-CIFAR10 ($step = 7$) | 71.30% | 72.30% | 73.30% |

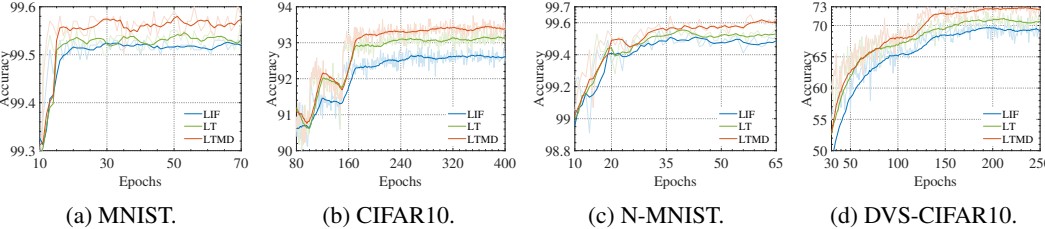

(a) MNIST.  (b) CIFAR10.  (c) N-MNIST.  (d) DVS-CIFAR10.

Figure 4: Inference accuracies with proposed methods implemented. Moving averages of 5 or 20 epoches are used in (a)/(c) and (b)/(d). Light-color curves are the original data.

The number of timesteps used in previous works and ours are compared in Tab. 2. It can be seen from the table that our SNNs use much fewer timesteps in training and inferencing compared to other SNNs (whether based on ANN-to-SNN conversion or backpropagation) and our proposed methods can significantly improve the performance of SNNs in many few timesteps. As a result, our proposed SNNs not only have a lower respond latency but also need less memory space due to the reduced temporal states of neurons.

## 4.2 Simulation Study

In this section, we conduct a series of comprehensive simulation studies to evaluate the effects of the proposed learnable thresholding and moderate dropout. We train SNNs with LIF and learnable thresholding neurons respectively, followed by integrating moderate dropout. Then, we test and compare the proposed SNNs' inference capabilities. As can be observed from Tab. 3 and Fig. 4, accuracies and convergence speed improve after using learnable thresholding, and are further enhanced with moderate dropout.

We show the change in network inference result during training to evaluate the effect of different initial threshold values on our network performance. Fig. 5 shows the high robustness of our proposed SNNs that comprise learnable thresholding neurons.

To analyze how thresholds change during network convergence, in Fig. 6, we use an 8-layer fully-connected SNN with 4 different initial thresholds to classify CIFAR10 dataset and plot the threshold update curves of neurons in different layers during training. It can be seen from the figure that although different initial threshold values are used, the neurons' threshold curves tend to converge after training with learnable thresholding. This shows the robustness of SNNs with learnable thresholding implemented to the initial threshold setting. Additionally, it can be seen that threshold

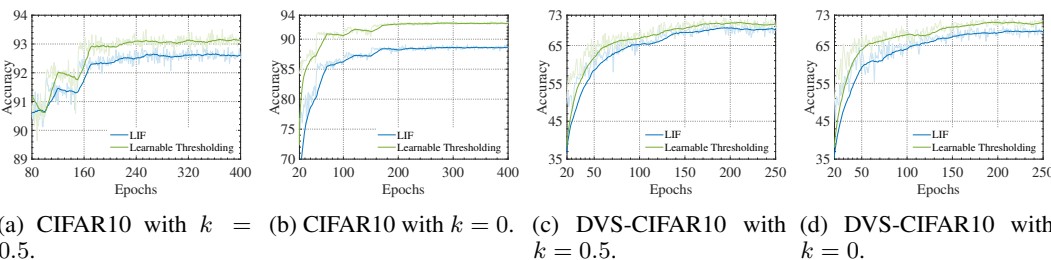

(a) CIFAR10 with $k = 0.5$.  (b) CIFAR10 with $k = 0$.  (c) DVS-CIFAR10 with $k = 0.5$.  (d) DVS-CIFAR10 with $k = 0$.

Figure 5: Inference accuracies of LIF and learnable thresholding on different datasets with various initial thresholds.

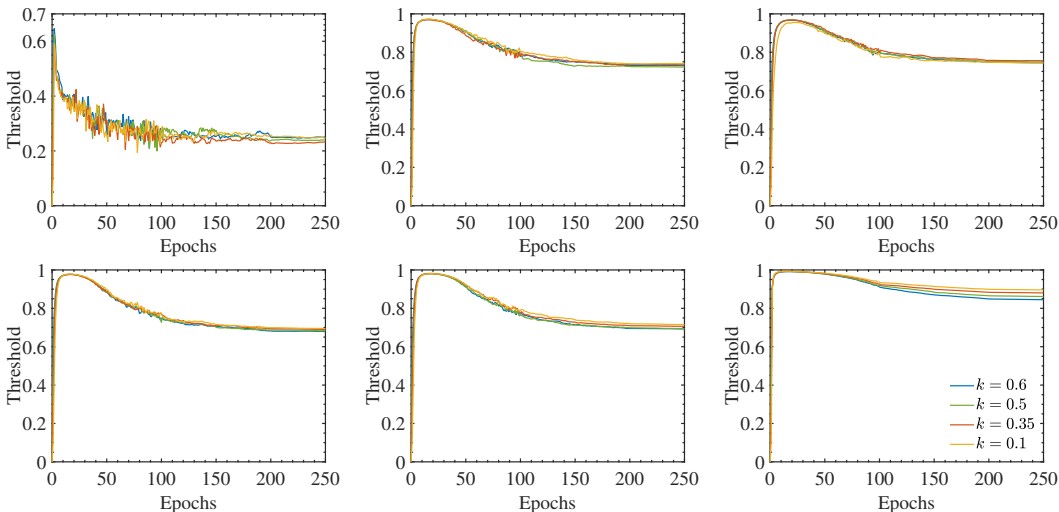

Figure 6: Thresholds change of different layers during network training with different initial threshold parameter $k$.

Table 4: Performance evaluation of LTMD with ResNet and VGG architectures.

| SNN architecture | LIF | LT | LTMD |
|---|---|---|---|
| ResNet19 | 91.97% | 93.59% | 93.78% |
| VGG16 | 91.68% | 92.10% | 92.35% |

values increase as the network goes deeper, indicating that information can pass through shallow-layer neurons to maintain sufficient data flow and be filtered by deep-layer neurons to extract useful information for classification.

The proposed learnable thresholding scheme is based on gradient descent method to update neuronal thresholds, which is always functionable when the activation gradient exists among network layers. And the moderate dropout scheme is a network stability enhancement strategy to be implemented to replace the original normal dropout in the model. Consequently, these proposed methods are not restricted to SNNs based on DenseNet architecture. We further conduct ablation study to evaluate effect of the proposed LTMD on network performance with ResNet and VGG architectures. The experimental results are shown in Tab. 4. Consistent performance improvement triggered by the proposed methods can be observed from this table, which verifies the generalized effectiveness of LTMD.

## 5 Conclusion

In this work, we propose learnable thresholding to endow spiking neurons with the ability of self-optimizing their threshold values during training and moderate dropout to enhance model stability by minimizing inconsistencies between output probability distributions in different sub-model runs. We construct deep SNNs based on DenseNet architecture and incorporate the aforementioned methods and obtaining promising results. Our proposed SNNs achieve higher accuracies for almost all the test datasets with significantly fewer timesteps required, which reflect their excellent classification capability and low system latency. We demonstrate higher robustness and faster convergence to the initial threshold for the learnable thresholding mechanism compared to using normal LIF neurons. The proposed methods' generalized effectiveness is also illustrated. Last but not least, we show that SNNs with learnable thresholding neurons have higher thresholds in deep layers after training to constraint the number of firing neurons, which proves the neuronal heterogeneity and helps to reduce the amount of computation.

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
