# A Appendix

## A.1 Algorithms

In this section, we provide the pseudo code of the potential-dependent dropping scheme (Algorithm 1) and the overall training procedures (Algorithm 2) of SNNs with our proposed methods.

---
**Algorithm 1** Potential-dependent dropping

---
**Input:** drop rate $p_{drop}$, input $X$.
**Output:** dropped input $o$ to the next layer.
**Parameter:** number of neurons $L(n)$ in layer $n$.
**Training:**
1: $u_{max} \leftarrow max(X)$
2: $u_{mean} \leftarrow average(X)$
3: **for** $i = 1$ to $L(n)$ **do**
4:    $u_i \leftarrow average(X_i(t))$
5:    $p_i \leftarrow p_{drop} \cdot \frac{u_{max} - u_i}{u_{max} - u_{mean}}$
6:    $mask_i = Bernoulli(p_i)$
7:    **for** $t = 1$ to $T$ **do**
8:       $o_i(t) \leftarrow mask_i \cdot X_i(t)$
9:    **end for**
10: **end for**
11: **return** $o$

---

## A.2 Details of Datasets and Training Settings

### A.2.1 MNIST

The MNIST dataset contains 60000 images for training and 10000 for testing. Each sample in MNIST is a gray-scale handwritten digit in size of $28 \times 28$ pixels.

### A.2.2 CIFAR10

The CIFAR10 is a collection of 60000 color images, divided into 50000 images for training and 10000 images for testing. All images are equally distributed and labelled as 10 classes. Samples in CIFAR10 are color images of size $32 \times 32$ and contain 3 RGB channels.

### A.2.3 N-MNIST

It is the neuromorphic transformed version of MNIST dataset, composed of event data stored in 2 spike trains per pixel. A saccade moving Dynamic Version Sensor (DVS) is used to capture intensity change of MNIST pixels displayed on an LCD screen. Intensity increment and decrement are saved as ON and OFF events in the dataset. Due to the sensor saccade moving, images recorded are slightly larger than the original MNIST images, in size of $34 \times 34 \times 2$. Each spike train in N-MNIST lasts $300000\mu s$, and we first reduce the resolution by 1000 times and then divide it into 15 steps. Each step contains the accumulated events within a period of $20ms$ in our experiment.

### A.2.4 DVS-CIFAR10

DVS-CIFAR10 is converted from its static version CIFAR10. There are 10 classes with 1000 images each in DVS-CIFAR10. The dataset image consists of $128 \times 128 \times 2$ spike trains and very noisy background, make it challenging for recognition tasks. We reduce resolution of DVS-CIFAR10 by 1000 times and accumulate events every $20ms$.

### A.2.5 Loss Function and Optimizer

For all the experiments except N-MNIST in this work, we use cross entropy and stochastic gradient descent (SGD) optimizer with momentum of 0.9 to calculate the error and iteratively update the synapses. Specially for N-MNIST dataset, Adam and mean squared error (MSE) are used.

**Algorithm 2** Overall training algorithm

---

**Input:** input $X$, target label vector $Y$.
**Output:** layer $i$ synaptic weights $W_i$, threshold $M_i$, prediction output $P, Q$.
**Parameter:** total number of network layers $N$, connections within receptive field $W_{rf}$.
**function** $NeuronUpdate(u, X, i)$
  1: **for** $t = 1$ to $T$ **do**
  2:    $u(t) = \kappa u(t-1) + W_i X(t)$
  3: **end for**
  4: **if** $u(t) \geq M_i$ **then**
  5:    $o(t) = 1$
  6:    $u(t) = 0$
  7: **else**
  8:    $o(t) = 0$
  9: **end if**
10: **return** $o$
**function** $FCUpdate(u, X, i)$
11: **for** $t = 1$ to $T$ **do**
12:    $u(t) = \kappa u(t-1) + W_i X(t) + W_{rf} o_i(t)$
13: **end for**
14: **if** $u(t) \geq M_i$ **then**
15:    $o(t) = u(t)$
16:    $u(t) = 0$
17: **else**
18:    $o(t) = 0$
19: **end if**
20: **return** $o$
**Training:**
21: **Forward:**
22: $o_1 \leftarrow NeuronUpdate(u_1, X, i = 1)$
23: **for** $i = 2$ to $N - 2$ **do**
24:    $o_i \leftarrow NeuronUpdate(u_i, o_{i-1}, i)$
25: **end for**
26: $o_{N-2} \leftarrow concatenate(o_{N-2})$
27: $o_{N-1} \leftarrow FCUpdate(u_{N-1}, o_{N-2}, N-1)$
28: $o_{N-1} \leftarrow PotentialDependentDrop(p_{drop}, o_{N-1})$
29: $o_N \leftarrow FCUpdate(u_N, o_{N-1}, N)$
30: $P, Q \leftarrow average(deconcatenate(o_N(t)))$
31: $E \leftarrow Loss(P, Q, Y)$
32: **Backward:**
33: $\frac{\partial \mathcal{L}}{\partial W}, \frac{\partial \mathcal{L}}{\partial k} \leftarrow Autograd$
**Inference:**
34: $o_1 \leftarrow NeuronUpdate(u_1, X, i = 1)$
35: **for** $i = 2$ to $N - 2$ **do**
36:    $o_i \leftarrow NeuronUpdate(u_i, o_{i-1}, i)$
37: **end for**
38: **for** $i = N - 1$ to $N$ **do**
39:    $o_i \leftarrow FCUpdate(u_i, o_{i-1}, i)$
40: **end for**
41: $P \leftarrow average(deconcatenate(o_N(t)))$

---

### A.2.6 Neuron Setting

Neurons in this work share the same decay factor $\kappa = 0.25$ and have reset state of 0. As events in neuromorphic datasets are sparse, we set the overall dropping rate of moderate dropout $p_{drop} = 0.2$ for all the experiments so that sufficient spiking neurons can be retained for information processing.

### A.2.7 Network Setting

*Down sampling* layers are used to reduce the feature map size and set with initial *output channel* = 256 for static datasets and 64 for neuromorphic datasets. *Conv pooling* layer with *kernel size* = 3 and *stride* = 2 is used to reduce the feature map size while maintaining sufficient information of the original samples. One more *Avg pooling* layer is used for DVS-CIFAR10 dataset due to its large sample size. All *Conv* in dense blocks and transition layers have *stride* = 1, *padding* = 1, and corresponding kernel sizes indicated in Fig. 2. *Avg pooling* layers in the SNN are configured to have *kernel size* = 2 and *stride* = 2 for reducing feature map sizes. *Adaptive avg pooling* layer can ensure output feature is of size $1 \times 1$ to make it manageable for computation. Two $FC$ layers further reduce the '*in feature*' to 256 and eventually $C$, where $C$ is the total number of candidate classes. Lateral interaction effect with receptive field width $d_{rf} = 5$ is considered in both $FC$ layers. Our proposed moderate dropout is placed in between the $FC$ layers. In the end, the last output decoding layer averages the accumulated output potentials followed by the softmax function to give the final prediction.

### A.2.8 Computational Resource

Network models in this work are programmed in Pytroch. All the experiments were conducted on NVIDIA GeForce RTX 2080 Ti. Some models were trained on multiple GPUs. Detailed information about number of GPUs we used and batch size for each experiment trial are listed in Tab. 5.

Table 5: Computational resource in experiments.

| Model | Batch size | GPU | Total iterations |
|---|---|---|---|
| MNIST ($step = 4$) | 80 | 2 | 70 |
| CIFAR10 ($step = 2$) | 64 | 2 | 400 |
| CIFAR10 ($step = 4$) | 64 | 4 | 400 |
| N-MNIST ($step = 15$) | 40 | 4 | 65 |
| DVS-CIFAR10 ($step = 7$) | 40 | 4 | 250 |

### A.3 Multiple-trial Experiment Results

Multiple-trial experiments have been conducted and results are reported in Tab. 6, showing the consistent high performance of our model.

Table 6: Accuracies of LTMD model for different datasets. All values are calculated based on 5 trials.

| Model | Best record | Mean | Standard deviation |
|---|---|---|---|
| MNIST ($step = 4$) | 99.60% | 99.584% | 0.012% |
| CIFAR10 ($step = 4$) | 94.19% | 94.154% | 0.0403% |
| N-MNIST ($step = 15$) | 99.65% | 99.614% | 0.0206% |
| DVS-CIFAR10 ($step = 7$) | 73.30% | 72.92% | 0.319% |

### A.4 Computational Load of SNN

In Tab. 7, we analyze the training time, number of additions, number of multiplications, and number of learnable parameters of our SNN on CIFAR10 using 4 timesteps. For ANN, an addition and a multiplication are combined as multiply-accumulate (MAC) operation and we also provide these counts in the table. In SNNs, neurons' outputs are binarized and only one addition operation is needed if neuron fires. The additions count for SNN is calculated by $r \times T \times A$, where $r$ is the average firing rate, $T$ is the total timesteps, and $A$ is the addition count in ANN. Average firing rates of neurons in each layer of SNN with LTMD applied are shown in Fig. 7. It should be noted that our SNN

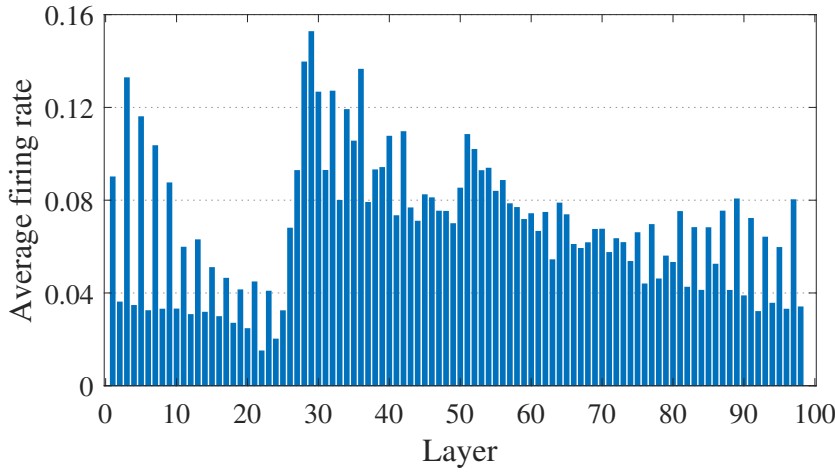

Figure 7: Average firing rate of neurons in each layer.

need multiplications because the encoding layer encodes real-value input into spikes and decoding layer uses neuronal potentials to provide network predictions. The comparison to ANNs indicates that event-driven SNNs can have significant operation reduction. And the proposed LTMD will only induce affordable computational load to the system but effectively enhance the performance.

Table 7: The computational load and learnable parameters of SNNs. Percentage increased based on static thresholding is shown in brackets.

| Computational load | Static threshold | LT | LTMD | ANN |
|---|---|---|---|---|
| Training time | 5762.92min | 5955.76min (+3.35%) | 6086.49min (+5.61%) | |
| Learnable parameters | 6.82M | 6.82M (+0.0015%) | 6.82M (+0.0015%) | |
| #Additions | 196.83M | 214.81M (+9.14%) | 212.55M (+7.99%) | 648.46M |
| #Multiplications | 28.85M | 28.85M (+0%) | 28.86M (+0.035%) | 648.46M |

### A.5 Noise Tolerance of Lateral Interactions

To investigate system robustness, we measure normal output decoding (without lateral interaction) and our model's performance on test dataset that is appended with Gaussian noise. Tab. 8 shows the inference accuracies on MNIST test datasets with different level of noise, mean value ranges from 0 to 0.5 with an interval of 0.1.

Table 8: SNN performance under different noise levels.

| Model | Noise=0 | Noise=0.1 | Noise=0.2 | Noise=0.3 | Noise=0.4 | Noise=0.5 |
|---|---|---|---|---|---|---|
| SNN | 99.53% | 90.60% | 82.33% | 75.76% | 54.16% | 31.56% |
| SNN&LI | 99.60% | 96.54% | 92.10% | 83.61% | 69.05% | 35.79% |

From the table we can conclude that our decoding layer with lateral interaction outperforms the one without lateral interaction at all noise level situations, especially when noise interference becomes large.