# OpenReview forum: "LTMD: Learning Improvement of Spiking Neural Networks with Learnable Thresholding Neurons and Moderate Dropout"
_NeurIPS.cc/2022/Conference — NeurIPS 2022 Accept_

### Official Review · Reviewer_n9SC · 2022-07-04

**Rating:** 6
**Confidence:** 3
**Soundness:** 3 good
**Presentation:** 3 good
**Contribution:** 3 good

**Summary:**

In this work, the authors inspired by the neuroscience observation that different neuronal mechanisms exist in disparate brain regions to propose a new spiking neuronal mechanism, named dynamic thresholding. It is clear that the goal of this study is to mimic the brain's ability to build different types of neurons that facilitate the transmission of information in a deep spiking model. Overall, this work is interesting.

---------------------------------------------------------------
The author has addressed all my concerns and I would like to upgrade my rating.

**Questions:**

1. One of my main concerns is that the author gives a github link in the abstract that provides the code, but the anonymous link to github should have this generated (i.e., https://anonymous.4open.science/). I am not sure if this issue violates the double-blind policy. If it does not violate it, I think this article should be WEAK ACCEPT.
2. The authors should explain the problem that the proposed model on the N-MNIST dataset in Table 2 has a slightly improvement in both performance and timesteps.

**Limitations:**

The authors have clarified.

**Strengths And Weaknesses:**

Strengths:
1. The new dynamic thresholding mechanism.
2. Moderate random deactivation technique based on DenseNet structure is used to enhance SNN modeling capability.

Weakness:
1. Comparison and analysis with other dynamic thresholding techniques is lacking.
2. The authors claim that output decoding can enhance the robustness of the system, but there are no experiments related to robustness in the manuscript. The reviewer suggests that if possible the authors need to simply supplement the SNR based input noise experiments to verify the robustness of the system. Unless the robustness claimed by the authors is not in the traditional sense.
3. The moderate dropout uses KL fused into the loss function, but the cross-entropy loss of the classification task already includes this component. The reviewer suggests that  the authors elaborate further.

---

> ### Author Response · Authors · 2022-08-02
> **Response to Reviewer n9SC (part 1/2)**
>
> Thank you for your insightful and very detailed feedback. We would like to address your concerns and answer your questions in the following.
>
> Q1: Comparison and analysis with other dynamic thresholding techniques is lacking.
>
> A1: Thank you for pointing this. Your concerns are the same with Reviewer BGk9. Therefore, we give the comparison in the General Response section and will add comparison description in the next version of the paper.
>
> ---
>
> Q2: The authors claim that output decoding can enhance the robustness of the system, but there are no experiments related to robustness in the manuscript. The reviewer suggests that if possible the authors need to simply supplement the SNR based input noise experiments to verify the robustness of the system. Unless the robustness claimed by the authors is not in the traditional sense.
>
> A2: Thanks for the feedback. To investigate system robustness, we measure normal output decoding (without lateral interaction) and our model’s performance on test dataset that is appended with Gaussian noise. Below table shows the inference accuracies on MNIST test datasets with different levels of noise, mean value ranges from 0 to 0.5 with an interval of 0.1. For this validation test, we use $T$=4.
>
> | Model  | Noise=0 | Noise=0.1 | Noise=0.2 | Noise=0.3 | Noise=0.4 | Noise=0.5 |
> |:------:|:-------:|:---------:|:---------:|:---------:|:---------:|:---------:|
> |  SNN   | 99.53%  |  90.60%   |  82.33%   |  75.76%   |  54.16%   |  31.56%   |
> | SNN&LI | 99.60%  |  96.54%   |  92.10%   |  83.61%   |  69.05%   |  35.79%   |
>
> **Tab. RT5: SNN performance under different noise levels.**
>
> From Tab. RT5 we can conclude that our decoding layer with lateral interaction outperforms the one without lateral interaction at all noise level situations, especially when noise interference becomes large. We will add this ablation study on noise interference in the next version of the paper.
>
> ---
>
> Q3: The moderate dropout uses KL fused into the loss function, but the cross-entropy loss of the classification task already includes this component. The reviewer suggests that the authors elaborate further.
>
> A3: The cross-entropy loss and KL divergence involved in the proposed loss function are two separate components and have different purposes. Cross-entropy is used to measure the error of actual output compared to target values in current iteration, which is exactly the same as the commonly used cross-entropy loss function in SNNs. But the KL divergence term is the key of the proposed moderate dropout method. Since both dropped neurons’ spatial and temporal connections will be omitted, normal dropout in SNN will lead to sub-models that have high unpredictable deviation. In such case, inference results of sub-models could be highly different, further causes model instability and affects model convergence. Moderate dropout is proposed to stabilize the SNN by minimizing the inconsistences between sub-models’ outputs. And the inconsistences are measured by KL divergence of the sub-models’ outputs. In this situation, we respectively take the actual output of each sub-model as the “target” distribution and calculate KL divergence between other sub-model’s output and it. In this part, we only care about the inconsistences among sub-models’ output distributions but not the correctness of these actual output. The additional KL term in the proposed loss function emphases and weighted scales (via multiplying by coefficient $\gamma$) the effect of sub-model inconsistences so that such inconsistences can be reduced effectively as network convergence.
>
> ---
>
> Q4: One of my main concerns is that the author gives a github link in the abstract that provides the code, but the anonymous link to github should have this generated (i.e., https://anonymous.4open.science/). I am not sure if this issue violates the double-blind policy. If it does not violate it, I think this article should be WEAK ACCEPT.
>
> A4: We checked the policy before submission, there is no requirement on github link format (normal or anonymous). To avoid policy violation, the link we use in the abstract is a newly created account with random name, and we are sure that no identifying information contained in either code or account. So it should not violate any double-blind policy. Thank you.

---

> ### Author Response · Authors · 2022-08-02
> **Response to Reviewer n9SC (part 2/2)**
>
> Q5: The authors should explain the problem that the proposed model on the N-MNIST dataset in Table 2 has a slightly improvement in both performance and timesteps.
>
> A5: For static datasets, constant real-value inputs are fed into the SNN at each timestep, whereas for dynamic datasets, events within certain period are used as the sequential inputs to the SNN. Consequently, for dynamic datasets whose events are changing over time, more timesteps are generally required to cover more input information. Each N-MNIST sample contains 34×34×2 spike trains and lasts 300ms, which are smaller and shorter than those in DVS-CIFAR10 (128×128×2) and make it computational affordable for us to utilize all events in the dataset. Therefore, most current SNNs [R11, R12] utilize the entire N-MNIST spike trains in their models. In our experiment shown in Tab. 2, we divide the spike train into 15 timesteps, each timestep contains accumulated events within the interval of 20ms. Same as other works, we utilize the whole spike train 15×20=300ms. Therefore, timesteps used on N-MNIST is slightly larger than our SNNs applied on other datasets. Due to DTMD methods proposed, our model achieves better performance as reported in Tab. 2.
>
> [R11] Guo, Yufei, et al. "RecDis-SNN: Rectifying Membrane Potential Distribution for Directly Training Spiking Neural Networks." Proceedings of the IEEE/CVF Conference on Computer Vision and Pattern Recognition. 2022.
>
> [R12] Wu, Yujie, et al. "Direct training for spiking neural networks: Faster, larger, better." Proceedings of the AAAI Conference on Artificial Intelligence. Vol. 33. No. 01. 2019.

---

### Official Review · Reviewer_khhY · 2022-07-08

**Rating:** 5
**Confidence:** 4
**Soundness:** 3 good
**Presentation:** 4 excellent
**Contribution:** 3 good

**Summary:**

The authors use dynamic thresholding to incorporate heterogeneity in the neuronal parameters of spiking neurons. Dynamic thresholding utilizes learnable threshold values and enables flexible neuronal mechanisms across layers with proper information flow within the network, and fast network convergence. The authors also propose a moderate dropout method to serve as an enhancement technique to minimize inconsistencies between independent dropout runs. The authors show that these methods increase the performance and robustness of the model for image classification.

**Questions:**

Since the authors introduce a dynamic thresholding technique, it would be great if the authors could show the increase in computational load due to this.

Also, a comparative study on the training times for the DTMD with the static thresholding case would be also helpful. A table showing the percentage change in performance on complex datasets with the increase in training time/computational load (measured using MACs/FLOPs)

The illustration showing the KL divergence of different runs of SNN sub-models is a bit unclear and not properly explained in the main text. It would be great if the authors could furnish some more details. Especially, the authors are claiming the output of the spiking neural network as a probability distribution, on which they are taking the KL divergence which seems unclear.

The authors use the “KL divergence between the different output distributions from sub-network runs is incorporated into the loss function according to Eq. 12, with the overall objective being to minimize the loss, leading to increased consistencies among different runs of the network.” However, the authors add the two KL divergences to get the overall loss. However, this is still not a metric as it still does not satisfy the triangle inequality i.e., it does not measure the distance between the two distributions. Thus, I believe effectively trying to minimize the symmetrized KL divergence counteracts and nullifies each other. It would be great if the authors could show an ablation study with and without using the symmetrized KL divergence and using each of the asymmetric KL divergences also.


**Limitations:**

The authors did not discuss the limitations of their work or the potential negative social impact of their work in the paper.

**Strengths And Weaknesses:**

Strengths:
The paper addresses a key problem that plagues current SNN research - i.e., homogeneity in the neuronal parameters. Though some recent papers have shown increased performance and robustness when heterogeneity is introduced in the neuronal parameters, the process of hyperparameter optimization is complex especially when the number of neurons is large. This paper uses a backpropagation-based method to find the optimal hyperparameters for dynamic thresholding which is great.

Though the performance of the model is promising, there are some key weaknesses that caught my attention.
Weakness:
The concept of introducing heterogeneity is not novel as several recent papers have introduced the concept and shown great results.
The paper shows results on small datasets like MNIST, CIFAR10, NMNIST - similar results have already been shown in recent related works.
The authors introduce two very different concepts - dynamic thresholding and moderate dropout which seems to be two independent modifications both of which are supposed to greatly increase the computational load

---

> ### Author Response · Authors · 2022-08-02
> **Response to Reviewer khhY (part 1/3)**
>
> Thank you for your very detailed comments and suggestions for improvement. We would like to address your concerns and answer your questions in the following.
>
> Q1: Since the authors introduce a dynamic thresholding technique, it would be great if the authors could show the increase in computational load due to this.
>
> Also, a comparative study on the training times for the DTMD with the static thresholding case would be also helpful. A table showing the percentage change in performance on complex datasets with the increase in training time/computational load (measured using MACs/FLOPs)
>
> A1: Thank you for the insightful suggestion. We conduct an ablation study here using our SNN on CIFAR10 dataset with 4 timesteps to measure model computational load. Because addition dominates the calculation in SNNs, we count number of additions and multiplications separately instead of using MACs in Tab. RT3. We use the same calculation method as [R6], the addition count is calculated by $r×T×A$, where $r$ is the average firing rate, $T$ is the timestep, $A$ is the addition count in the original ANN. And the multiplication count in SNN still exists because the first encoding layer converts real values to spikes and decoding layers use potentials to predict output class. Increment percentages based on static thresholding are shown inside brackets. In this table, we also list addition and multiplication count for ANNs to demonstrate SNNs’ computation reduction property.
>
> |  Computational load  | Static threshold |         DT          |        DTMD         |  ANN  |
> |:--------------------:|:----------------:|:-------------------:|:-------------------:|:-----:|
> |    Training time     |    5762.92min    | 5955.76min (+3.35%) | 6086.49min (+5.61%) |       |
> | Learnable parameters |      6.82M       |  6.82M (+0.0015%)   |  6.82M (+0.0015%)   |       |
> |     \#Additions      |     871.47M      |  951.11M (+9.14%)   |  941.08M (+7.99%)   | 2900M |
> |  \#Multiplications   |      28.85M      |    28.85M (+0%)     |  28.86M (+0.035%)   | 2900M |
>
> **Tab. RT3: The computational load and learnable parameters of SNNs. Percentage
> increased based on static thresholding is shown in brackets.**
>
> From the table, number of learnable parameters increased due to dynamic thresholding is very low and can be neglected since one threshold value will be shared by neurons in the same layer. The proposed moderate dropout will only be implemented in fully-connected output layers, therefore, only a bit more multiplications will be induced during training. Extra addition operations triggered by the proposed DTMD is around 8%. As computational load brought by the proposed methods is manageable, only 3-5% extra training time is needed.
>
> [R6] Li, Yuhang, et al. "Differentiable spike: Rethinking gradient-descent for training spiking neural networks." Advances in Neural Information Processing Systems 34 (2021): 23426-23439.
>
> ---
>
> Q2: The illustration showing the KL divergence of different runs of SNN sub-models is a bit unclear and not properly explained in the main text. It would be great if the authors could furnish some more details. Especially, the authors are claiming the output of the spiking neural network as a probability distribution, on which they are taking the KL divergence which seems unclear.
>
> A2: Thanks for pointing this. Noting that different neurons will be randomly selected to be retained or dropped from the network during training, each individual run of the network will lead to inputs interacting with different sub-models of the network, especially for spiking neurons who have additional temporal connections. Consequently, results between different runs may deviate unpredictably, and affect the network inference. In our SNN, the last-layer neurons’ average potentials over all timesteps are used as the output, which can be treated as the probabilities of classifying input pattern to each potential class after softmax. Different sub-models generate different output potentials combinations, which can be treated as different probability distributions. KL divergence is a statistical distance to measure the differences between two probability distributions by its definition. As such, we select KL divergence as the measurement of inconsistencies between output probability distributions generated by different sub-models running over the same mini-batch, and then, minimize such inconsistencies to enhance network stability. In Fig. 3, two rows represent two sub-models’ architectures after dropout. Because spiking neurons are spatio-temporally connected, each sub-model can be unfolded into multiple timesteps $T$, represented by $T$ squares in the figure. Due to the different combinations of neurons remaining after dropout, these sub-models may output different results. $P$ and $Q$ are the time-averaged output distributions of the two sub-models and will be used to calculate KL divergence. We will polish the description in the next version of the paper.

---

> ### Author Response · Authors · 2022-08-02
> **Response to Reviewer khhY (part 2/3)**
>
> Q3: The authors use the “KL divergence between the different output distributions from sub-network runs is incorporated into the loss function according to Eq. 12, with the overall objective being to minimize the loss, leading to increased consistencies among different runs of the network.” However, the authors add the two KL divergences to get the overall loss. However, this is still not a metric as it still does not satisfy the triangle inequality i.e., it does not measure the distance between the two distributions. Thus, I believe effectively trying to minimize the symmetrized KL divergence counteracts and nullifies each other. It would be great if the authors could show an ablation study with and without using the symmetrized KL divergence and using each of the asymmetric KL divergences also.
>
> A3: Thanks for your suggestion. In Eq. 12, we add the two KL divergences to make it symmetric. We assume that if the two asymmetric KL divergences are in great value difference, symmetry would be important for robustness improvement because using the smaller valued asymmetric KL divergence will reduce the impact on total loss function. For example, if $P(A)=0.5$, $P(B)=0.5$, $Q(A)=0.9$, $Q(B)=0.1$, then, $KL(P\parallel Q)=0.51$, $KL(Q\parallel P)=0.37$. $KL(P\parallel Q)$ is about 38% higher than $KL(Q\parallel P)$ in value.
>
> We agree with the reviewer that even summing the two KL divergences still does not satisfy the triangle inequality, so it is not a metric. But the symmetrized KL divergence will not counteract each other because no term in the equation $KL(P\parallel Q)+KL(Q\parallel P)=\sum_{x\in\mathcal{X}} P(x)\log(\frac{P(x)}{Q(x)})+\sum_{x\in \mathcal{X}} Q(x)\log(\frac{Q(x)}{P(x)})$ could be cancelled out. Thank you for your advice on testing asymmetric KL divergence instead, results of ablation study are shown below.
>
> |                  | Without $KL$ | $KL(P\parallel Q)+KL(Q\parallel P)$ | $KL(P\parallel Q)$ | $KL(Q\parallel P)$ |
> |:----------------:|:------------:|:-----------------------------------:|:------------------:|:------------------:|
> | Inference result |    92.88%    |               93.75%                |       93.66%       |       93.62%       |
>
> **Tab. RT4: Effect of asymmetric KL divergence.**
>
> ---
>
> Q4: The authors did not discuss the limitations of their work or the potential negative social impact of their work in the paper.
>
> A4: Our proposed methods will only add on several (same as number of network layers) learnable parameters, which is negligible compared to the millions of learnable synaptic weights in an SNN as mentioned in line 162. Computational load will be increased a bit after implementing the proposed methods. We will add computational load table in the next version of the paper.
>
> Our methods are performance enhancement strategies for direct trained SNNs, which do not have any negative social impact. It has been indicated in the checklist.

---

> ### Author Response · Authors · 2022-08-02
> **Response to Reviewer khhY (part 3/3)**
>
> Q5: The concept of introducing heterogeneity is not novel as several recent papers have introduced the concept and shown great results. The paper shows results on small datasets like MNIST, CIFAR10, NMNIST - similar results have already been shown in recent related works.
>
> A5: The aim of our work is not to show that neuronal heterogeneity is a critical property for spiking neurons and can benefit SNN performance, some other works [R7, R8] have already introduced that. In this work, we want to show a practical threshold optimization methodology to enhance neuronal heterogeneity to those who are already interested in this special property of SNNs and expecting an easy implementation. With that said, we briefly introduce the information capturing and noise tolerance effect of various threshold levels in line 132-136, and graphically illustrate diverse neuronal outputs caused by different thresholds in Fig. 1b. In addition, we provide threshold levels distribution of trained network layers in Fig. 6 and give an explanation from information processing’s perspective in line 310-312, that experimentally matches neuronal heterogeneity concept.
>
> As shown in Tab. 2 and ablation study of applying DTMD on other network architectures in the General Response section, our proposed methods can consistently provide promising results, and more importantly, without much computational load increment as shown in Tab. RT3 in our response A1. Compared to other recent works where spike state conversion [R9] or approximated target states distribution [R10] are additionally required for algorithm deployment, the proposed methods can be directly implemented on any BPTT-based SNN. That can greatly simplify the algorithm migration.
>
> [R7] Rathi, Nitin, and Kaushik Roy. "Diet-snn: Direct input encoding with leakage and threshold optimization in deep spiking neural networks." arXiv preprint arXiv:2008.03658 (2020).
>
> [R8] Fang, Wei, et al. "Incorporating learnable membrane time constant to enhance learning of spiking neural networks." Proceedings of the IEEE/CVF International Conference on Computer Vision. 2021.
>
> [R9] Guo, Yufei, et al. "RecDis-SNN: Rectifying Membrane Potential Distribution for Directly Training Spiking Neural Networks." Proceedings of the IEEE/CVF Conference on Computer Vision and Pattern Recognition. 2022.
>
> [R10] Meng, Qingyan, et al. "Training High-Performance Low-Latency Spiking Neural Networks by Differentiation on Spike Representation." Proceedings of the IEEE/CVF Conference on Computer Vision and Pattern Recognition. 2022.

---

> ### Author Response · Authors · 2022-08-07
> **Response to Reviewer khhY – A Statement of Computational Load Increment**
>
> Thank you for your interest in our backpropagation-based threshold updating scheme especially for networks of large number of neurons. Here we want to re-elucidate the computational load increment triggered by the proposed methods.
>
> The proposed dynamic thresholding scheme is designed on the basis of backpropagation, making the algorithm can be merged to any SNN that is directly trained by gradient descent. Resultant increment of learnable parameters is negligible compared to network synaptic connections since all neurons in the same layer share the same threshold value. As illustrated in the table in our rebuttal, addition and multiplication counts do not increase much after implementing our methods. This is because both dynamic thresholding and moderate dropout will only take effect in network training stage, but not inference. Therefore, all the methods will not largely affect the forward-path calculation during inferencing. We only observe very small change in average firing rate after utilizing our methods, leads to small computational load increment as shown in the ablation study. It is noted that compared to the counterpart ANN, great addition and multiplication operations deduction still can be observed in our SNN after DTMD implemented. Considering the generalized model performance enhancement, we think such increasing load is affordable.
>
> Based on the above explanation, we believe that the computational load triggered by our methods is clearly stated and can address your concerns. We understand your intensive review procedure during this very tight discussion period, but we really hope you can reconsider your rating. Thank you again and look forward to your further comments and suggestions.

---

> > ### Comment · Reviewer_khhY · 2022-08-08
> > **Thanks**
> >
> > I believe the authors have sufficiently answered the questions I had. Though the results are very encouraging, I still don't feel there are sufficiently new things in the paper, as the main contribution of the paper, which is the dynamic thresholding and moderate dropout, seems to be previously published, as the authors pointed out. However, owing to the extensive experiments and detailed analysis, I would stick to my initial rating.
> >
> > Good luck!

---

> > > ### Author Response · Authors · 2022-08-09
> > > **Thanks for Your Reply and Clarification on the “New Things”**
> > >
> > > Thank you for your reply and affirmation. But we still want to re-clarify the “new things” in this paper. We have never said that our proposed method, either dynamic thresholding or moderate dropout, has been published previously. Dynamic thresholding is proposed to enhance heterogeneity of neurons to improve SNN performance. It is true that our method is not the first one that brings the concept of heterogeneity to spiking neurons. But our method provides a feasible and computational efficient way to effectively optimize neurons’ thresholds as mentioned in General Response section, which makes it a distinct approach. More importantly, our dynamic thresholding scheme can be migrated to any gradient descent-based SNN to enhance network performance, leading to a much wider application domain than previous work. Moderate dropout is proposed to stabilize SNN through minimizing inconsistences between sub-models. To the best of our knowledge, such an idea has never been mentioned in any previous work.
> > >
> > > Thanks again for your suggestions and affirmation.

---

### Official Review · Reviewer_BGk9 · 2022-07-11

**Rating:** 4
**Confidence:** 5
**Soundness:** 2 fair
**Presentation:** 2 fair
**Contribution:** 2 fair

**Summary:**

The authors proposed a learnable (authors called dynamic) threshold scheme and a moderate dropout method for SNNs. The effectiveness
of the proposed approach is reflected by classification tasks and DenseNet. Based on the experimental results, the proposed method requires fewer time steps for achieving state-of-the-art classification performance.

**Questions:**

Please see weaknesses.

**Limitations:**

1. The proposed approach is not a dynamic threshold scheme. I suggest the authors call it a learnable threshold scheme.
2. The authors should compare other existing dynamic threshold schemes to the proposed learnable threshold, e.g., r1 and r2.
3. The validation only based on DesneNet is very limited. The authors should provide more experimental results to show the effectiveness of the proposed approach.


r1: Yunzhe Hao, Xuhui Huang, Meng Dong, and Bo Xu. A biologically plausible supervised learning method for spiking neural networks using the symmetric stdp rule. Neural Networks, 121:387–395, 2020.
r2: TaeyoonKim,SumanHu,JaewookKim,JoonYoungKwak,JongkilPark,SuyounLee,InhoKim,Jong- Keuk Park, and YeonJoo Jeong. Spiking neural network (snn) with memristor synapses having non-linear weight update. Frontiers in computational neuroscience, 15:22, 2021.

**Strengths And Weaknesses:**

Strengths:

1. Looking into authentic bio-plausible dynamic threshold schemes is a promising research direction for SNNs.
2. Even though the authors did not develop an actual dynamic threshold scheme, assigning different thresholds to different SNN neurons is interesting.

Weaknesses:

-1. If I understand correctly, the proposed threshold scheme is not dynamically changed during inference. But, it is just learnable parameters during a training process. If so, I do not think it is a "dynamic" threshold scheme, which is just a learnable threshold scheme.

-2. The validation mainly relies on DesneNet. But the authors used 8-layer FC SNN for MNIST. How does the DesneNet work with the MNIST dataset? Does the proposed method work with other architectures well, e.g., ResNet, VGG?

-3. lines 282: "much fewer time steps needed" does not mean "suitable for processing real-time data." It depends on how much time spend on each time step. The logic does not hold here. The authors should report the processing speed for each time step.

-4. There are so many "-" in table 2. The authors did not explain why which I think is very necessary.

-5. The authors claimed "less memory space" but without providing corresponding proofs.

In summary, I think the validation is not convincing. If the proposed learnable threshold scheme only works with DenseNet, it is just an application. To make it more compelling, the authors should show the generalization of the proposed method. In addition, the authors should also show the experimental results of other tasks besides classification tasks. However, based on the current manuscript, I cannot see any generality of the proposed approach.

---

> ### Author Response · Authors · 2022-08-02
> **Response to Reviewer BGk9 (part 1/2)**
>
> Thank you for your constructive feedback. Please see our detailed reply below.
>
> Q1: If I understand correctly, the proposed threshold scheme is not dynamically changed during inference. But, it is just learnable parameters during a training process. If so, I do not think it is a "dynamic" threshold scheme, which is just a learnable threshold scheme.
>
> A1: We agree with the reviewer that our proposed method is to grant the learning capability to neuronal thresholds in each layer during training stage, so that thresholds can automatically approach their optimal values specifically for current task and model architecture as network convergence. And apply the trained threshold values during inference to enhance heterogeneity of SNN neurons. By applying this thresholding scheme, selecting proper threshold value to avoid extreme neuronal firing probabilities when constructing SNNs will no longer be an issue. The main advantage of our method is that it can be easily implemented and enhance SNN performance without as much computational load or slow convergence as with the STDP-based dynamic threshold schemes, especially when number of neurons becomes very large.
>
> Since most recent high-performance SNNs use a pre-defined “fixed” threshold value throughout both training and inference, as a counterpart, we named our method as “dynamic” thresholding to emphasize the self-updating ability of neurons’ thresholds. But we are always open to any more appropriate name.
>
> ---
>
> Q2: The validation mainly relies on DesneNet. But the authors used 8-layer FC SNN for MNIST. How does the DesneNet work with the MNIST dataset? Does the proposed method work with other architectures well, e.g., ResNet, VGG?
>
> A2: Compared with the other three datasets, MNIST is a relatively simple one in terms of image size, channel numbers, and noise involved. Therefore, even the 8-layer SNN can achieve 99.60% classification accuracy on this dataset. Our DenseNet-based SNN can provide very high nonlinearity so that theoretically more suitable for complex datasets. It is mentioned in [R4] that some datasets are too simple and even not applicable for deep SNNs. However, our SNN still works on MNIST. By applying DenseNet-based SNN on MNIST, 99.58% accuracy can be achieved. Although the result is very close to the one reported using the 8-layer SNN (99.60%), the training computational load becomes much larger (total training time: 481.29min vs 174.34min; memory space: 17340MiB vs 11154MiB) due to the complex network structure. Considering the accuracy, time consumption, and memory occupancy, we choose to use non-DenseNet model to present the result in the paper. Another reason for creating this 8-layer FC SNN is that we want to demonstrate the effectiveness of our methods on different architectural SNNs.
>
> Our proposed methods are not DenseNet-specific and can work well with other architectures. Please see our ablation experiment results in general response. Thank you!
>
> [R4] Zheng, Hanle, et al. "Going deeper with directly-trained larger spiking neural networks." Proceedings of the AAAI Conference on Artificial Intelligence. Vol. 35. No. 12. 2021.
>
> ---
>
> Q3: lines 282: "much fewer time steps needed" does not mean "suitable for processing real-time data." It depends on how much time spend on each time step. The logic does not hold here. The authors should report the processing speed for each time step.
>
> A3: Thank you for your insightful comments. We agree that “fewer timesteps” does not equal to “real-time”. We conduct the ablation study on RTX 2080Ti using the trained model presented in Tab. 2 and record the total inference time for 10000 MNIST images is 6.8025s. We are using an SNN model whose timestep is set to 4, therefore, each timestep is about 0.17ms for processing 1 image.
>
> ---
>
> Q4: There are so many "-" in table 2. The authors did not explain why which I think is very necessary.
>
> A4: In Tab. 2, we want to compare our model with other state-of-the-art works. “-” in the table simply means not applicable or the result not reported by the original authors. As the ANN-to-SNN conversion methods are inherently not able to deal with temporal data (not applicable to dynamic datasets) and researchers tend to verify their models on different datasets, lack of record for certain model on certain dataset is quite common. And some SNN authors do not open source their code, we are not able to accurately recreate the model for evaluation. Therefore, in Tab. 2, we only present results that have been published.

---

> ### Author Response · Authors · 2022-08-02
> **Response to Reviewer BGk9 (part 2/2)**
>
> Q5: The authors claimed "less memory space" but without providing corresponding proofs.
>
> A5: Thank you for pointing this. Spiking neurons are spatio-temporally connected. When we use backpropagation through time (BPTT) method to directly train the SNN, all synaptic connections and neurons’ states need to be stored in memory for subsequent neurons’ states calculation and parameters update according to Eq. 2, 6, 7. We show that our SNN can achieve higher accuracies with fewer timesteps in Tab. 2, which means our SNN will be thinner after temporally unfolding than other works. Consequently, there will be fewer synaptic connections and neurons’ states need to be written in the memory. Here we conduct an experiment to record memory space occupancy by our SNN when classifying CIFAR10 using different timesteps.
>
> |     Timestep     |  $T$=2   |  $T$=3   |  $T$=4   |  $T$=5   |  $T$=6   |
> |:----------------:|:--------:|:--------:|:--------:|:--------:|:--------:|
> | Memory occupancy | 13852MiB | 19394MiB | 24606MiB | 30224MiB | 35948MiB |
>
> **Tab. RT2: Memory occupancy of the proposed model with different timesteps.**
>
> Trade-off between memory occupancy and model performance always exist in SNNs. Due to the superior performance, in order to achieve the same accuracy, our model generally needs fewer timesteps so that less memory space is needed. As a comparison, 93.16% inference result claimed in [R5] with $T$=6 and ResNet19 architecture consumes 26136MiB memory space. While our SNN ($T$=2) can get better result (93.75%) with only 53% memory space needed.
>
> [R5] Zheng, H., Y. Wu, L. Deng, et al. Going deeper with directly-trained larger spiking neural networks. arXiv preprint arXiv:2011.05280, 2020.
>
> ---
>
> Q6: The authors should compare other existing dynamic threshold schemes to the proposed learnable threshold.
>
> A6: Thanks for the constructive suggestion. Your suggestion is the same as Reviewer n9SC. Therefore, we provide the comparison in General Response section and will add it into the paper.

---

> ### Author Response · Authors · 2022-08-07
> **Response to Reviewer BGk9 – A Clarification on Generalized Effectiveness and Contribution of Our Methods**
>
> Thanks for your constructive discussion on generality of our proposed methods, and your time in reading our paper and rebuttal. Here we want to clarify a bit more on the generalized effectiveness of our proposed methods and contribution of this work.
>
> - The two methods proposed in this work, dynamic thresholding and moderate dropout are both independent on the network architecture utilized. The dynamic thresholding scheme is integrated with the gradient descent algorithm to update neuronal thresholds during training, which means the proposed thresholding scheme is always functionable when the SNN is trained by backpropagation through time (BPTT) method. In addition, moderate dropout will only replace the original dropout that have been widely utilized in SNNs, which is not restricted to any specific network architecture. To experimentally demonstrate the generalized effectiveness, we use two network architectures in the main content (DenseNet and full-connected) and add two more architectures (ResNet and VGG) in our rebuttal. Consistent improvement caused by our proposed methods can be observed from those ablation study results.
>
> - One of our main contributions is to propose a way to integrate neuronal threshold optimization with synaptic weights updating so as to enhance SNN heterogeneity without much computational load incurred. Because our thresholding scheme is 100% based on BPTT algorithm, it can be easily migrated to any other backpropagation-based SNNs without barrier. In addition, compared to the previous STDP-based threshold updating algorithms, whose hyperparameters updating process is complex especially when number of neurons becomes large, our scheme can find the optimal values of hyperparameters easily relies on the power of backpropagation.
>
> Based on these two points, we really hope that you can refer to our rebuttals and reconsider your judgement. On the other hand, we do not mean we already made everything detailed and clear in the paper, we will continue to polish descriptions in the future versions. We are happy to discuss further if you have any concerns.

---

> ### Comment · Reviewer_BGk9 · 2022-08-09
> **Thanks for the responds**
>
> Thanks for the additional experimental results and more details related to my questions!
>
> 1. The proposed method is not a dynamic threshold, especially since many dynamic threshold schemes have been proposed and tested. The proposed method is a learnable threshold, specifically learned for a specific task. Therefore, calling the proposed approach a "dynamic threshold" is misleading.
>
> 2. The real dynamic threshold change threshold is based on the input during inferences, meaning it could deal with degraded conditions or the conditions differ from the training data. I am not convinced the proposed approach could deal with those conditions as the threshold is fixed.
>
> 3. Even though the authors provided the experimental results with VGG and ResNet, I think more thorough experiments and analyses should be conducted.
>
> Therefore, I will keep my original rate.

---

### Author Response · Authors · 2022-08-02
**General Response**

We thank all reviewers for their time and constructive feedback. In this general response, we would like to address the concerns about comparison to other dynamic threshold schemes and our proposed methods’ effectiveness on other network architectures.

**Comparison to other dynamic threshold schemes.**

We agree that several dynamic thresholding schemes have been developed in recent years to make neuronal thresholds learnable and vary across the network. However, most of the existing methods require grim prerequisites or not based on gradient descent methodology, resulting in higher computational load or are difficult to be migrated to other SNNs. In [R1], spike threshold is trainable to minimize the quantization error if all neurons’ states are represented by the proposed scaled weighted firing rate. Pre-defined target firing count is compulsory and neurons’ actual firing count need to be recorded periodically to adjust neuronal thresholds in [R2]. STDP-based dynamic threshold scheme used in [R3] can only modify excitatory neurons’ threshold values to balance neuronal firing rates, which leads to slow convergence and heavy computational load because every neuron has different firing history and holds different threshold values. Compared to all these proposed dynamic threshold methods, ours can be utilized on any backpropagation-based direct trained SNN without hindrance and does not induce large training parameters increment. We would like to clarify that our threshold learning method is proposed to be directly integrated with the existing backpropagation-based training algorithms so that all neuronal parameters, including threshold, can self-optimize during training without any prerequisite or incurring large computational load. We will add the comparison to other threshold schemes in the next version of the paper.

[R1] Meng, Qingyan, et al. "Training High-Performance Low-Latency Spiking Neural Networks by Differentiation on Spike Representation." Proceedings of the IEEE/CVF Conference on Computer Vision and Pattern Recognition. 2022.

[R2] Kim, Taeyoon, et al. "Spiking neural network (snn) with memristor synapses having non-linear weight update." Frontiers in computational neuroscience 15 (2021): 646125.

[R3] Hao, Yunzhe, et al. "A biologically plausible supervised learning method for spiking neural networks using the symmetric STDP rule." Neural Networks 121 (2020): 387-395.

**The proposed methods’ effectiveness on other architectures.**

Our methods’ effectiveness is independent of the network architectures used. Our proposed thresholding scheme iteratively updates neurons’ thresholds based on backpropagation, which is always functionable as long as gradients exist among layers. And the moderate dropout scheme is proposed to replace the common dropout applied in a network, which is also not restricted to any specific architecture.

In the paper, we have tested our methods on both DenseNet based and 8-layer full-connected SNNs. We evaluate our DTMD with ResNet and VGG networks for further support. The experiments are conducted on CIFAR10 dataset with timestep $T$=2, all results are shown in below table.

| SNN architecture |  LIF   |   DT   |  DTMD  |
|:----------------:|:------:|:------:|:------:|
|     ResNet19     | 91.97% | 93.59% | 93.78% |
|      VGG16       | 91.68% | 92.10% | 92.35% |

**Tab. RT1: Performance evaluation of DTMD with DenseNet and VGG architectures.**

Consistent performance improvement triggered by the proposed methods can be observed from this table, which verifies the generalized effectiveness of DTMD. We would like to note that although ResNet can achieve similar results as DenseNet, more learnable parameters are needed (ResNet: 12.63M vs DenseNet: 6.82M). We will add generalized effectiveness analysis in our next version of the paper.

---

### Public Comment · ~Armstrong_The_Second1 · 2023-02-06
**What a great writing skill that fools all reviewers, including AC! !**

A prior work PLIF [r1] is cited in this paper and clearly achieved better performance on MNIST, CIFAR10, and DVS-CIFAR10. But, the authors neither list PLIF results in the main experiment (Table 2) nor explain the reason why not. What's worse, the authors still claim that they have achieved the SOTAs.

All three reviewers did not notice this issue.  **Should I call this dishonest writing or tricky writing?  That is a question!**

**I recommend anyone who has read my comment be cautious of this work. The contribution is very limited, and the ablation study of the proposed tricks is missing. Most importantly, this work may be suspected of dishonest writing.**





[r1] Fang, W., Yu, Z., Chen, Y., Masquelier, T., Huang, T., & Tian, Y. (2021). Incorporating learnable membrane time constant to enhance learning of spiking neural networks. In Proceedings of the IEEE/CVF International Conference on Computer Vision (pp. 2661-2671).

---

> ### Public Comment · Authors · 2023-02-07
> **Irresponsible and unprofessional remarks**
>
> It is sad that such individual who supposedly is learned in the field can resort to making such unprofessional remarks.  Quite clearly, this individual is not interested in looking or reading a paper and evaluate the merit of the work in its totality.  It is very easy to take random potshots and make irresponsible remarks that has no basis.  Whatever the agenda this person has, I can only guess!
> Nevertheless, not worth spending time to give a technical rebuttal because the remark is not even a question that deserves an answer.
>
> If indeed the interest is for technical advancement, I do suggest to openly direct technical questions to me specifically.

---

> > ### Public Comment · ~Armstrong_The_Second1 · 2023-02-07
> > **It's not lies that cut but the sharpness of the truth. Humiliating me does not cover up the truth of the mistake.**
> >
> > If you were innocent of my accusation, why did you insult me rather than confront me? If you were confident in your honesty, why did you stigmatize me rather than justify yourself? If you were professional enough, show me your defense rather than guess my intention. If you were responsible for the field, you should mend the problem in your paper rather than look down on anyone with any reasonable doubt.
> >
> > Escaping from answering me is the positive answer to your dishonest issue. What are you scared of? The revealed truth? What made you so panic and rude? The hurtful truth?
> >
> > Actually, I read not only your paper from beginning to end but also all three reviewers' comments. **And, I am very certain that my charge for your dishonest issue is very concrete.**
> >
> > For one thing, you claim about three times that LTMD could achieve superior results over prior works (Last sentence of the abstract, last sentence of the introduction, and the last fourth sentence of Page 8), which proves the emphasis/highlight of the paper is on performance improvement. For another, [r2] uses the same techniques as [r1] but has lower accuracy due to a different model architecture. [r2] is listed in your main experiment (table 2) because your model can win over them, while [r1] that greatly beats your work is not listed.  Clearly, you did not compare your work with the real SOTAs you know, given [r1] is also cited in the paper, and **you lied about the significance of your main contribution**.
> >
> > You did not do sufficient experiments as [r1] did in the main experiment. Your ablation study on MD is missing. I am fine with it since all three reviewers did not raise it. **But I am not tolerant of Dishonest or Cheating.** It is academic ethics that I question you for rather than any technical results.
> >
> > **Notably, I accuse you with evidence while you insult me with nonsense.** Anonymity is not for random potshots, but for protection from injustice. I still hold my stand for the insistence of doing no wrong.
> >
> >
> >
> >
> >
> >
> >
> > [r1] Fang, W., Yu, Z., Chen, Y., Masquelier, T., Huang, T., & Tian, Y. (2021). Incorporating learnable membrane time constant to enhance learning of spiking neural networks. In Proceedings of the IEEE/CVF International Conference on Computer Vision (pp. 2661-2671).
> >
> > [r2] Fang, W., Z. Yu, Y. Chen, et al. Deep residual learning in spiking neural networks. Advances in Neural
> > Information Processing Systems, 34, 2021.

---

> > > ### Public Comment · Authors · 2023-02-08
> > > **Not interested to comment yet**
> > >
> > > Nope, still not interested to give a technical rebuttal/explanations yet.
> > > I will reply provided... (see hint)
> > >
> > >
> > >
> > >
> > >
> > >
> > >
> > > Hint: basic social/professional etiquette and courtesy.

---

> > > > ### Public Comment · ~Armstrong_The_Second1 · 2023-02-08
> > > > **Not interested to follow up at all. Please don't be full of yourselves and self-obsessed. This is quite embarrassing. ^ ^!**
> > > >
> > > > Be clear. I am not requesting any technical explanations at all. As I commented at first, this work's contribution is very limited and is suspected of dishonest writing. The authors' choice to insult personally instead of justifying further confirmed the fact of the dishonest issue.
> > > >
> > > > **I believe the masses have sharp eyes.** I just need to alert anyone who could possibly visit this web. We shall see the follow-up to the paper. (including but not limited to, the committee's decision, citation, GitHub/codes influence, and community recognition)
> > > >
> > > > I will not do further comments anymore. That will be waste of time. I've done enough.

---

> > > > > ### Public Comment · Authors · 2023-02-08
> > > > > **Thank you**
> > > > >
> > > > > Thank you.

---

> > ### Public Comment · ~Wei_Fang2 · 2023-02-24
> > **Suggesions to resolve the argument**
> >
> > Hi, I am the first author of [r1]. I just noticed the violent discussion about this paper.
> >
> > This paper achieves higher accuracy on CIFAR-10 but lower accuracy on CIFAR10-DVS than [r1]. However, I also notice that this paper uses less time-steps than [r1]. Both high accuracy and small time-steps are desirable in SNNs.
> >
> > To resolve the argument, I suggest that the authors of this paper should be careful about writing. For example, the authors can claim that they achieve SOTA accuracy when compared with some certain researches, rather than say that they get the highest accuracy.

---

### Meta-Review · Area_Chair_pFs6 · 2022-08-24

**Recommendation:** Accept
**Confidence:** Less certain

**Metareview:**

This paper introduces an trainable threshold and a dropout variant to improve training of spiking neural networks. There were serious issues raised by the reviewers primarily about 1) the idea of having trainable spiking thresholds being not new, and 2) the paper providing insufficient computational validation. The authors addressed the second point. About the first point, the authors provided the following: "The aim of our work is not to show that neuronal heterogeneity is a critical property for spiking neurons and can benefit SNN performance, some other works [R7, R8] have already introduced that. In this work, we want to show a practical threshold optimization methodology to enhance neuronal heterogeneity to those who are already interested in this special property of SNNs and expecting an easy implementation." The AC agrees that the paper achieves this goal, and, after much thought, agreed that this is a sufficient contribution for acceptance to NeurIPS.

In addition, as a minor point, the AC agrees with the opinion of Reviewer BGk9 that the term "dynamic thresholding" is misleading.

**Award:**

No

---

> ### Public Comment · ~Armstrong_The_Second1 · 2023-02-06
> **Alert on Dishonest Issue!**
>
> I know this may be useless. But, I insist on alerting AC that this work may have the issue of dishonest writing. Please see my comment
> above.

---

### Decision · Program_Chairs · 2022-09-14

Accept

---

> ### Public Comment · ~Armstrong_The_Second1 · 2023-02-06
> **Alert on Dishonest Issue!**
>
> I know this may be useless. But, I insist on alerting PC that this work may have the issue of dishonest writing. Please see my comment above.